# Feasibility and Preliminary Effects of Acupuncture for Cognitive Dysfunction in Diverse Cancer Survivors: A Pilot, Randomized, Placebo-Controlled Trial

**DOI:** 10.3390/curroncol32010027

**Published:** 2025-01-01

**Authors:** Xiaotong Li, Kaitlin Lampson, Tim A. Ahles, James C. Root, Q. Susan Li, Yuelin Li, Anam Ahsan, Jun J. Mao, Kevin T. Liou

**Affiliations:** 1Department of Medicine, Memorial Sloan Kettering Cancer Center, New York, NY 10065, USA; lix9@mskcc.org (X.L.); lampsonk@mskcc.org (K.L.); liq2@mskcc.org (Q.S.L.); maoj@mskcc.org (J.J.M.); 2Department of Psychiatry and Behavioral Sciences, Memorial Sloan Kettering Cancer Center, New York, NY 10065, USA; ahlest@mskcc.org (T.A.A.); rootj@mskcc.org (J.C.R.); liy12@mskcc.org (Y.L.); anam.ahsann@gmail.com (A.A.); 3Department of Epidemiology and Biostatistics, Memorial Sloan Kettering Cancer Center, New York, NY 10065, USA

**Keywords:** acupuncture, pilot study, cognitive dysfunction, cancer survivorship, feasibility

## Abstract

(1) Background: This pilot study evaluates the feasibility and preliminary effects of acupuncture for cancer-related cognitive dysfunction (CRCD) in cancer survivors. (2) Methods: A randomized trial comparing real acupuncture (RA) to sham acupuncture (SA) and waitlist control (WLC) among cancer survivors reporting cognitive difficulties. Interventions were delivered weekly over 10 weeks. Feasibility was evaluated by recruitment, treatment adherence, and assessment completion. Subjective CRCD was assessed by the Functional Assessment of Cancer Therapy-Cognitive Function—Perceived Cognitive Impairment subscale (FACT-Cog PCI) and objective CRCD was assessed by the Hopkins Verbal Learning Test—Revised (HVLT-R). (3) Results: 32 participants (57.1% of eligible patients) were enrolled. All participants in acupuncture groups completed ≥8 of 10 treatments. Assessment completion rate was 100% for all participants. From baseline to week 10, the RA group (n = 19) reported a clinically meaningful 17.3-point increase in FACT-Cog PCI (95% confidence interval [CI] 12.5 to 22.1), compared to 9.7 points (95% CI 2.8 to 16.7) in the SA group (n = 9), and 6.8 points (95% CI −3.7 to 17.2) in the WLC group (n = 4). In the subgroup analysis among patients with a below-average baseline HVLT-R (T-score < 50), the RA group (n = 8) increased FACT-Cog PCI scores by 20.4 (95% CI 13.6 to 27.3), compared to 11.1 points (95% CI 0.6 to 21.5) in the SA group (n = 5). The improvements from RA persisted through week 16 in both the total sample and the sub-group. Eleven mild adverse events were reported, with pain and bleeding at the needling sites being the most common. (4) Conclusions: The findings support the feasibility and safety of conducting a randomized, placebo-controlled trial to evaluate acupuncture for cognitive dysfunction in cancer survivors.

## 1. Introduction

Cancer-related cognitive dysfunction (CRCD) describes a range of cognitive impairments associated with cancer and its treatments, including disruptions in memory, attention, learning, language, and processing speed [1,2,3]. CRCD affects up to 75% of cancer patients in active treatment and 46% of survivors after treatment [4,5], posing significant challenges to their self-care and daily lives [3,6,7,8,9]. Impaired cognitive function has been shown to significantly predict difficulties in medication management, such as forgetting to take medications, even when controlling for sociodemographic and clinical confounders [10]. Furthermore, CRCD hinders survivors’ ability to return to work, impacts social relationships, erodes self-confidence, and in some cases, leads to early retirement [11]. Although National Comprehensive Cancer Network (NCCN) guidelines have recommended non-pharmacological treatments such as cognitive rehabilitation, behavioral therapy, and physical activity for CRCD [12,13,14], the evidence base for these therapies is still limited [15,16,17,18,19,20,21,22]. As a result, a standard of care has yet to be established [12,13], underscoring the urgent need to develop novel non-pharmacological interventions for CRCD.

Acupuncture has been widely used in cancer centers across the U.S. and shown potential for treating CRCD [23]. In a meta-analysis investigating the benefits of acupuncture for managing side effects induced by drug therapies in patients with breast cancer, four studies focused on cognitive impairment, measured by both subjective and objective assessments, demonstrated a significant therapeutic effect of acupuncture compared to sham acupuncture, no treatment, or waitlist controls (pooled SMD = −0.57; 95% CI [−0.96, −0.17]; *p* = 0.005; I^2^ = 89%) [24]. Additionally, in a secondary analysis (n = 99) from a large clinical trial involving cancer survivors with insomnia, acupuncture produced significant within-group improvements in subjective CRCD, as well as objective measures of attention (Cohen’s d = 0.29), learning (Cohen’s d = 0.31), and memory (Cohen’s d = 0.33) that persisted 12 weeks post-treatment [25]. However, existing studies are primarily limited to white or Asian populations, post-chemotherapy breast cancer patients, lack sham control groups, or are secondary analyses. Therefore, more studies with rigorous design are needed to investigate the efficacy of acupuncture on CRCD in a broader and more diverse cancer population compared to sham controls.

Thus, we designed this Cancer-related Cognitive Function Acupuncture Pilot Study (CLARITY) to evaluate the feasibility, safety, and preliminary effects of acupuncture for CRCD in a diverse cancer population. Given that acupuncture for CRCD has not been directly explored in a three-arm trial with survivors of diverse cancer types, feasibility is the primary aim of this trial. Piloting study procedures is especially important, considering the distinct challenges that cancer populations with CRCD face—such as high symptom burden and the additional mental demands of treatment and assessment schedules, compared to cancer patients without CRCD [26]. This feasibility study focused on testing recruitment procedures, defining and refining safe and replicable acupuncture and sham interventions, and evaluating outcome assessment burden for cancer survivors with CRCD [27]. As a secondary aim, we explored the preliminary effects of real acupuncture versus sham acupuncture and a waitlist control on both subjective and objective cognitive measures. Findings from this study will provide insights for the design of larger future clinical trials aimed at improving cognition in cancer populations.

## 2. Materials and Methods

### 2.1. Study Design

The CLARITY study is a three-arm, parallel, randomized controlled trial (RCT) to evaluate the feasibility, safety, and preliminary effects of real acupuncture (RA) versus sham acupuncture (SA) and waitlist control (WLC) for CRCD in cancer survivors. The study was conducted at Memorial Sloan Kettering Cancer Center locations throughout New York and New Jersey. Interventions were delivered over 10 weeks, and outcomes were assessed at weeks 0, 4, 10, and 16. Upon completion of the 16-week study, patients in the SA and WLC groups had the option of receiving 10 sessions of real acupuncture within the subsequent 6 months.

This study attempted to follow the Obesity-Related Behavioral Intervention Trials (ORBIT) model, which is a useful framework for developing and refining novel non-pharmacological interventions [27]. Given the paucity of studies that have been designed to assess acupuncture for CRCD, we identified this trial as aligning with the ORBIT Phase I, which promotes the design and refinement of a safe intervention in a population of interest to inform future trial design. Therefore, during the study period, we made several amendments to improve the feasibility and refine the study protocol for a future RCT (Appendix A). This study and all the amendments were approved by the institutional review board at Memorial Sloan Kettering Cancer Center (IRB number: 19-179).

### 2.2. Study Participants

English-speaking, adult participants were eligible if they (1) had a prior diagnosis of stage 0-III cancer; (2) completed initial cancer treatments (including surgery, chemotherapy, and/or radiation therapy) at least one month prior to study enrollment (patients receiving maintenance cancer treatment with hormonal or targeted therapies were permitted); (3) reported moderate or greater perceived cognitive impairment as indicated by a score of “quite a bit” or “very much” on at least one of the two items on the EORTC QLQ-C30 instrument (version 3.0) [28,29] that specifically address concentration (Item #20) and memory (Item #25); (4) indicated that their cognitive functions have worsened since their cancer diagnosis by replying “yes” to all of the following questions: “Do you think or feel that your memory or mental ability has gotten worse since your cancer diagnosis?”, “Do you think your mind isn’t as sharp now as it was before your cancer diagnosis?”, and “Do you feel like these problems have made it harder to function on your job or take care of things around the home?”; and (5) were willing to adhere to all study-related procedures, including randomization to one of three groups.

Participants were excluded if they (1) had active disease; (2) used acupuncture for cognitive symptom management within the 3 months before enrollment; (3) had a diagnosis of Alzheimer’s Disease, vascular dementia, Parkinson’s disease, or another organic brain disorder; (4) recorded a score of greater than or equal to 10 on the Blessed Orientation-Memory-Concentration (BOMC) screening instrument [30]; (5) had a diagnosis of a primary psychiatric disorder not in remission; or (6) had a change in somnogenic medication (e.g., hypnotics, sedatives, and/or antidepressants) in the four weeks prior to enrollment.

### 2.3. Procedures

After initial screening, patients who were deemed potentially eligible had a confirmatory eligibility visit with a clinician. Eligible patients provided informed consent before completing baseline assessments. Once baseline assessments were completed, participants were randomized in a 2:1:1 ratio to RA, SA, and WLC, respectively, stratified by prior chemotherapy use. Randomization was carried out by a secure computer system using a permuted block randomization protocol. Patients in the intervention groups remained blinded during the study, and they were notified of their treatment allocation once considered off-study. The research coordinators and biostatistician and data manager were blinded to treatment allocation. The principal investigator and other co-investigators were unblinded so that they can better understand the effects of the acupuncture interventions for the purpose of guiding intervention development and ensuring patient safety. Acupuncturists were unblinded as well because they were administering the RA and SA treatments.

It is important to note that the original study protocol planned to enroll 80 patients; however, recruitment stopped early due to the launch of a larger three-arm, parallel RCT designed to build on this early-phase study. Furthermore, the allocation of participants did not follow the original 2:1:1 ratio, as we removed the WLC group partway through the study to allocate more patients to the RA and SA groups, whose feasibility was less established in acupuncture trials compared to waitlist controls (Appendix A).

### 2.4. Interventions

#### 2.4.1. Real Acupuncture (RA)

RA is a therapeutic modality derived from Traditional Chinese Medicine (TCM), inserting thin, sterile, single-use, metallic needles into body surface with manual and/or electrical stimulation [31]. Acupuncturists had over 10 years of experience in clinical oncology environments and were trained on the procedure by the primary investigator. This study used a semi-fixed manualized protocol which consisted of a core set of points to address cognition and supplementary points tailored by acupuncturists to manage co-morbid symptoms such as insomnia, fatigue, and/or psychological distress (Appendix A). Point selection was based on input from acupuncturists with oncology expertise, consultation with experienced acupuncturists in China [32,33], and common points used in clinical trials to treat cognitive difficulties in the scientific literature. Acupuncturists were also allowed to use clinical judgment to add or remove up to 6 acupoints to or from the protocol for patient safety and comorbid symptoms. Any additional points were carefully documented, including the rationale for their use. Moreover, if clinically appropriate, the acupuncturist could apply electrostimulation to a maximum of four needles. The total number of needles used per session ranged from 10 to 26. Needles (0.20 or 0.25 × 30 or 40 mm, SEIRIN-America Inc., Weymouth, MA, USA) were inserted to appropriate depths and manipulated to achieve “De Qi”, a localized sensation of soreness, numbness, and/or distension around acupoints [34]. Needles were left in place for 20–30 min. Treatments were audited bi-weekly to ensure quality and adherence to treatment protocols.

#### 2.4.2. Sham Acupuncture (SA)

SA is a placebo acupuncture procedure. It was delivered as described in RA with the following differences: (1) Instead of inserting needles into core or supplementary points, acupuncturists chose non-acupuncture points that were not trigger points (Appendix A). (2) Instead of puncturing the skin with needles, the acupuncturists taped needles (0.20 or 0.25 × 30 or 40 mm, SEIRIN-America Inc., Weymouth, MA, USA) to the skin without any additional manual stimulation. The SA treatment was designed to give patients the same amount of personal attention from clinicians, identical treatment duration, and a similar overall experience to RA without meaningfully stimulating any real acupuncture points. After completion of the week 16 outcomes, patients in the SA group were eligible to redeem 10 free acupuncture treatments to be used in the subsequent 6 months.

#### 2.4.3. Waitlist Control (WLC)

Patients randomized to the WLC received standard care for their symptoms as prescribed by their healthcare providers. After completion of week 16 outcomes, patients in the WLC group were eligible to redeem 10 free acupuncture treatments to be used in the subsequent 6 months.

### 2.5. Outcomes

#### 2.5.1. Feasibility and Safety Outcomes

Limited research has been conducted in survivors of various cancer types to evaluate acupuncture for subjective and objective CRCD. Therefore, guided by methodological experts, we defined a priori quantitative benchmarks for the following three aspects of feasibility: recruitment, treatment adherence, and outcome completion. The research team determined that the study would be deemed feasible if 50% of eligible patients agreed to participate in the study and 75% of enrolled participants adhered to acupuncture treatments (completed at least 8 of the 10 treatment sessions) and outcome assessments. Recruitment, treatment adherence, and outcome completion for each patient were carefully recorded by research staff.

Safety was assessed by documenting adverse events across treatment arms and grading their severity using the Common Terminology Criteria for Adverse Events (CTCAE) v5.0 guidelines [35].

#### 2.5.2. Subjective Cognitive Function

The perceived cognitive impairment (PCI) subscale from the Functional Assessment of Cancer Therapy–Cognitive Function (FACT-Cog) version 3 was the primary outcome for assessing subjective CRCD. FACT-Cog is a self-reported questionnaire validated in cancer populations and consists of 37 items for evaluating patient’s cognitive function [36,37]. It has four domains: PCI, impact on quality of life, comments from others, and perceived cognitive abilities [37,38]. Items are rated on a 5-point Likert scale ranging from 0 “Never” or “Not at all” to 4 “Several times a day” or “Very much” in the previous seven days [36]. We chose to use the PCI subscale (Cronbach’s α 0.94) [39] of FACT-Cog as the primary outcome as this outcome is patient-centered [40] and consistent with the recommendations of the International Cognition and Cancer Task Force (ICCTF) [41]. The total scores of this subscale, calculated by summing the 18 items, range from 0 to 72, with higher scores indicating better cognitive function. According to the established minimal clinically important difference (7.4 points post treatment) [39], we conservatively defined a meaningful cognitive treatment change as a total PCI score increase ≥ 7.5. Patients completed the FACT-Cog questionnaire at Weeks 0, 4, 10, and 16.

#### 2.5.3. Objective Cognitive Function

The Hopkins Verbal Learning Test—Revised (HVLT-R) is one of the most widely used verbal memory assessments and has high test-retest reliability and validity and is recommended by the ICCTF for assessing cognition in cancer survivors [41,42,43]. The test consists of three trials of free recall of a 12-item list of words from three semantic categories. The ‘Total Recall’ score is the sum of the words recalled in each of the three learning trials (range: 0 to 36). The ‘Delayed Recall’ score is the number of list items the patient can recall after a 20–25-min delay period (range: 0–12). To control for practice effects, six different forms of the test were developed [44]; three of these were selected for the purposes of this trial and each was administered once per testing timepoint (baseline, week 10, and week 16). We used published norms to convert the raw score to an age-based T-score. In this study, we divided participants into two groups, below- (T < 50) and above-average cognitive function (T ≥ 50), based on their HVLT-R Delayed Recall T-score at baseline.

### 2.6. Statistical Analysis

Data were analyzed following intent-to-treat (ITT) principles. Descriptive statistics were used to summarize the feasibility outcomes as well as demographic and clinical characteristics (e.g., age, gender, and cancer type) at baseline.

To examine the mean change in FACT-Cog PCI scores in three groups from baseline to week 10 and 16, we employed a linear mixed-effects model [45]. We used a subject-specific random intercept to account for the correlation between repeated measures of the outcome. Fixed effects included treatment, time, treatment by time interaction, and baseline outcome. Given that the HVLT-R was a secondary outcome for this pilot study, we reported mean differences and confidence intervals, without testing for statistical significance. Both the FACT-Cog and HVLT-R outcomes were evaluated in the total sample and among participants with below-average objective function at baseline, defined as an HVLT-R T-score < 50.

All analyses were two-sided with a *p*-value of less than 0.05 for statistical significance. Statistical analyses were conducted using STATA (version 18.0; STATA Corporation, College Station, TX, USA) and SAS (version 9.4; SAS Institute, Inc., Cary, NC, USA).

## 3. Results

### 3.1. Participant Characteristics

From October of 2020 to March of 2021, we screened 116 cancer survivors, and excluded 84, of which 60 were deemed either ineligible (N = 31) or potentially eligible in the future (N = 29); 24 declined to participate or were unable to be contacted. A total of 32 participants were enrolled and randomized into three groups: RA (n = 19), SA (n = 9), and WLC (n = 4) groups. All participants completed their assigned treatment secession (completed at least eight out of 10 treatments) and follow-up assessments and were included in the final analysis (Figure 1).

Table 1 shows the demographic and clinical characteristics of the participants. The mean age was 58.0 years (standard deviation [SD], 11.9 years). The majority of participants were women (25, 78.1%), white (22, 68.7%), and not Hispanic or Latino (29, 90.6%). More than half of the participants (17, 53.1%) had an advanced education degree. Diverse cancer types were represented, the most common being breast (34.4%) and gynecological (25.0%). The most common cancer treatments in this population are surgery (84.4%) and chemotherapy (46.9%). The mean time since diagnosis was 5.7 years (SD, 10.8 years). The majority of patients were diagnosed with stage I (35.7%) and stage II (35.7%). The mean baseline FACT-Cog score was 35.6 (SD, 13.0). Participants in all three groups were similar in all characteristics.

### 3.2. Feasibility and Safety

Thirty-two participants (57.1% of eligible patients) were enrolled. All participants in acupuncture groups completed ≥8 of 10 treatments. Assessment completion rate was 100% for all participants. Thus, our feasibility benchmarks were met.

There were 11 unique reports of adverse events across the acupuncture arms. Pain and bleeding at the needling sites were the most common adverse events, followed by bruising, itchiness, leg cramping, and nausea. All adverse events were graded as ‘Mild’, or Grade 1, the lowest severity rating according to Common Terminology Criteria for Adverse Events (CTCAE) guidelines [35].

### 3.3. The Effects of Acupuncture on Subjective Cognition

From baseline to week 10, RA resulted in a clinically meaningful increase in FACT-Cog PCI by 17.3 points (95% CI 12.5 to 22.1), compared to 9.7 points (95% CI 2.8 to 16.7) in the SA group, and 6.8 points (95% CI −3.7 to 17.2) in the WLC group. At week 16, the effect of RA on FACT-Cog PCI remained stable, with an improvement of 17.3 points (95% CI 12.5 to 22.2). Improvements at week 16 in the FACT-Cog PCI were 11.1 points (95% CI 4.2 to 18.0) in the SA group and 12.3 points (95% CI 1.84 to 22.66) in the WLC group. However, the between-group differences were not statistically significant (Table 2) (Figure 2a).

We further conducted a restricted analysis among participants with below-average objective cognitive function (those with a baseline T-score < 50 on the HVLT-R). Since only one participant from the WLC group met this criterion, we focused on participants in the RA and SA groups. At week 10, participants in the RA group (n = 8) reported an increase in FACT-Cog PCI of 20.4 points (95% CI 13.6 to 27.3), compared to an increase of 11.1 points (95% CI (0.6 to 21.5) in the SA group (n = 5). By week 16, the improvements in the RA group remained stable at 20.7 points (95% CI 14.0 to 27.3), which was higher compared to the improvements in the SA group (7.3 points, 95% CI −3.3 to 17.9); the between-group difference was statistically significant (*p* = 0.044) (Table 2) (Figure 2b).

### 3.4. The Effects of Acupuncture on Objective Cognition

HVLT-R total and delayed recall for participants in all three groups at the different time points are summarized in Table 3. Similarly to the FACT-Cog PCI analysis, we conducted the analysis for HVLT-R among all participants and those with below-average baseline objective cognitive function. Due to the exploratory nature of the HVLT-R analysis, we present descriptive statistics without significance testing (*p*-values). When restricted to patients with a baseline HVLT-R T-score < 50, there was a potential trend of improvement in both HVLT-R total and delayed recall scores in the RA group at weeks 10 and 16 (Table 4). However, in the SA group, while participants showed some improvement at week 10, their scores declined by week 16, falling below baseline levels. Only one person in the WLC group met the criteria for the subgroup analysis, so they were excluded from these analyses.

## 4. Discussion

To our knowledge, this is the first study to evaluate the feasibility and safety of acupuncture for CRCD compared to sham and WLC controls in a survivorship population that included multiple cancer types. The results suggest that acupuncture is a safe and feasible intervention for CRCD in a research setting. Additionally, our results raise the question of whether acupuncture could produce clinically meaningful and persistent improvements in both subjective and objective CRCD compared to sham and WLC, particularly among patients with below-average objective cognitive function at baseline. These preliminary findings can be leveraged to design future, large-scale efficacy trials of acupuncture for CRCD.

Given the limited evidence for acupuncture in treating CRCD, a feasibility study is essential to minimize the risk of compromising the results of a larger RCT due to unanticipated challenges, such as trial design, recruitment strategies, or the effectiveness and acceptability of the intervention [46,47,48]. Following the ORBIT framework, this study is best positioned as a Phase I trial which seeks to define and refine a safe and feasible acupuncture intervention in cancer survivors [27]. Consistent with the early phase of this study, we made several adjustments, focusing on acupuncture protocols (e.g., session length, SA techniques, blinding), target population (e.g., inclusion and exclusion criteria, subgroup analysis plan), and research processes (e.g., flexible outcome collection methods) (details listed in Appendix A). Many of these changes were implemented to enhance recruitment efficiency and reduce the burden on patients participating in the research trial, thereby increasing treatment adherence and outcome completion. For example, we expanded inclusion criteria to encompass more cancer types, allowed for the remote administration of consent visits, implemented remote administration of the neurocognitive battery outcome, and expanded treatment locations to MSK regional sites. As a result, we exceeded our recruitment efficiency benchmark and achieved 100% treatment adherence and outcome collection for this study. This suggests that the current research plan is feasible and well-suited for advancing to the next phase of research.

Our study contributes to the growing body of research on non-pharmacological interventions for CRCD. Based on a literature search, we only identified three prior studies published in English that employ acupuncture for CRCD. Consistent with our preliminary findings, all of them reported positive effects of acupuncture on both subjective and objective cognitive impairments using various measures. However, these studies have limitations such as enrolling predominantly white or Asian populations, focusing solely on newly-diagnosed breast cancer patients actively receiving chemotherapy [49], being a single-arm pilot study (n = 12) [50], relying on secondary analysis [25], or lacking a sham control. Moreover, none of the three previous trials used both subjective and objective cognitive outcomes that are recommended by the ICCTF, which is a common limitation in this field that leads to difficulties in understanding the effects of interventions for CRCD in different contexts [51]. Our trial addressed these gaps by implementing a more rigorous design and using ICCTF-recommended outcomes. However, given that our study was not properly powered to detect effects, all preliminary results should be interpreted as hypothesis-generating.

Our study also contributes to the limited research on CRCD in minoritized racial/ethnic populations. Previous studies have shown that Black and non-white individuals with cancer are more likely to experience CRCD than their white counterparts, yet CRCD in minoritized racial and ethnic groups remains poorly understood [52]. A review indicated that most studies on CRCD management strategies have been conducted predominantly in white populations. This highlights the need to develop feasible strategies that promote the participation of minoritized racial and ethnic groups [3]. Although more inclusive representation is still needed, we were able to recruit a cohort of relatively diverse patients: 31.3% non-white; 9.4% Hispanic or Latino. The inclusion of patients from minoritized populations enhances the generalizability of the feasibility findings and is key to understanding how to address health disparities [53,54].

Our results raise the question of whether real acupuncture could produce more effective improvements in subjective cognition than both sham and WLC. Participants in the real group reported an increase of 17.3 points at both week 10 and 16, more than twice the clinically important difference established for this subscale (7.5 points), meeting the threshold criteria for a “much better” improvement as defined by Bell, et al. [39]. There were no statistically significant differences in most of the between-group comparisons, which are likely due to the small sample size and the large confidence intervals in both the SA and WLC groups. However, for both SA and WLC, the lower bounds of the confidence intervals fell below the threshold for clinically meaningful change, while the entire range of confidence intervals in the RA was above the threshold and narrower. This suggests more consistent effects within the RA group, with more patients achieving clinically meaningful improvements in cognition. The response observed in the SA group highlights the potential impact of treatment expectations and patient–provider relationships on patients’ perception of improvement. From a study design perspective, investigators should consider these patterns when calculating statistical power and determining an appropriate sample size for a future efficacy trial. The observed differences in confidence intervals suggest the potential superiority of RA over SA, but a larger efficacy trial is needed to confirm these preliminary findings.

We also included the objective cognitive measure HVLT-R. By offering patients the option to complete assessments virtually, along with other strategies to streamline the study procedures, we achieved a very high outcome completion rate compared to most intervention trials. The findings indicate that patients with below-average objective cognitive function at baseline showed greater improvements in both subjective and objective measures. Since CRCD is often closely associated with other physical and psychological symptoms such as anxiety, depression, and insomnia [2,3], it is possible that these comorbid symptoms may lead patients to believe they are experiencing cognitive decline subjectively. As a result, treatments targeting CRCD may not significantly address all comorbid symptoms, leading to less perceived improvement compared to those with below-average objective cognitive function. Moreover, patients reporting above-average cognitive function at baseline may have limited potential for improvement in their cognitive function after receiving treatments. These findings warrant further investigation.

Current treatment options for CRCD are limited. The recent NCCN guidelines recommend the use of non-pharmacological therapies for CRCD whenever possible, due to the higher risk of side effects and contraindications that current drug therapies pose [14]. Although many non-pharmacologic therapies such as physical exercise and cognitive rehabilitation for CRCD have been explored, a gold standard has yet to emerge. This is largely due to lacking or mixed evidence of efficacy and practical challenges in both research and clinical settings (e.g., low adherence [55] and limited access to qualified therapists [56,57]). Acupuncture is widely available in cancer centers [23] and well-tolerated by cancer survivors with high symptom burden [58,59,60,61]. Preliminary findings from this study suggest that acupuncture may have the potential to produce clinically meaningful and durable cognitive improvements, although this would have to be confirmed in a larger, adequately-powered RCT. While the mechanisms underlying these effects are not fully understood, basic science suggests that acupuncture may influence cognition via multiple neurobiological and physiological pathways [62], specifically by modulating signaling pathways involved in neuronal survival and function [63,64], mitigating central inflammation by suppressing oxidative stress [65], and enhancing neurotrophin signaling [64,66]. In patient populations, acupuncture has been shown to increase certain circulating proteins, hormones, and neurotransmitters that are involved with cognition [49,67,68]. Based on the emerging evidence, acupuncture may represent a practical, effective, and well-tolerated therapeutic option. Future trials should seek to further characterize the specific effects of acupuncture on various pathways involved in cognition to provide a framework for developing more targeted treatments.

Our study has several limitations. First, this study underwent several amendments related to study design during the study period, including the removal of a WLC group. Additionally, we stopped enrolling prior to reaching the target sample size due to the initiation of a larger, similar trial. These decisions reduce the statistical power to detect between-group differences. However, the primary purpure of this study was to design a safe, feasible, and replicable intervention that lays the foundation for future trials. We have made necessary changes for researchers to refine the study protocol. Therefore, the current preliminary effects should be viewed as a basis for the development of future trials and interpreted with caution regarding their clinical significance. Second, we did not collect information regarding other cognitive interventions that patients may have been using, such as mindfulness or exercise. Thus, we cannot confidently claim that acupuncture solely or even primarily caused the effects observed in this study. However, due to randomization, any use of outside interventions may be evenly distributed across study arms. Third, due to the small sample size, especially for the WLC group, larger trials may not have similar treatment adherence and outcome completion. Fourth, subjective and objective cognition are highly complex processes that can be measured in various ways, but in this study, we only assessed one subjective (FACT-Cog PCI) and one objective (HVLT-R) outcome measure. Additionally, we did not include measures for other comorbid symptoms, such as fatigue and sleep, which are highly correlated with cognitive deficits due to cancer [69]. Finally, our study was conducted in a large, urban, highly-resourced academic cancer center, where patients may have high adherence or other population-specific characteristics. Moreover, our study population was overrepresented by highly educated women, and the mean age of our patients is comparatively young for cancer survivors. Our findings cannot be generalized to other contexts or patient populations.

Our study also has many strengths. By directly addressing gaps in prior research, we demonstrated the feasibility and safety of acupuncture for CRCD in diverse cancer survivors. Future trials may seek to implement and further iterate on the acupuncture interventions that are outlined in this trial by testing them in a larger cohort of cancer survivors, and in different patient subpopulations and settings. Given the differing preliminary effects observed in patients with a baseline HVLT-R T-score below 50, future trials might consider stratifying participants based on baseline HVLT-R or other objective cognitive measures, or focusing solely on patients with low baseline scores. Moreover, future trials should seek to further test the durability of the treatment effects observed in this trial by implementing longer follow-up periods and more comprehensive subjective and objective measures.

## 5. Conclusions

In conclusion, our results support the feasibility and safety of a three-arm acupuncture trial to treat CRCD in diverse cancer survivors of various cancer types. The preliminary effects observed in the different treatment groups need to be confirmed in adequately powered efficacy trials. This study offers valuable insights for future trials utilizing non-pharmacological interventions to enhance cognitive function in cancer survivors.

## Figures and Tables

**Figure 1 curroncol-32-00027-f001:**
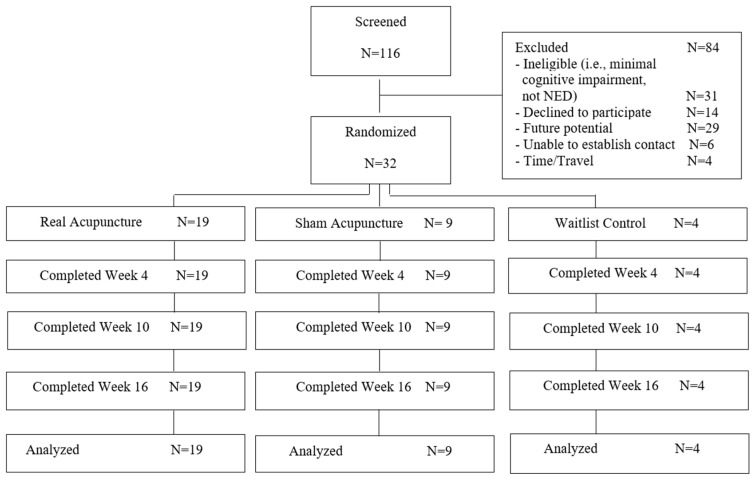
CONSORT diagram.

**Figure 2 curroncol-32-00027-f002:**
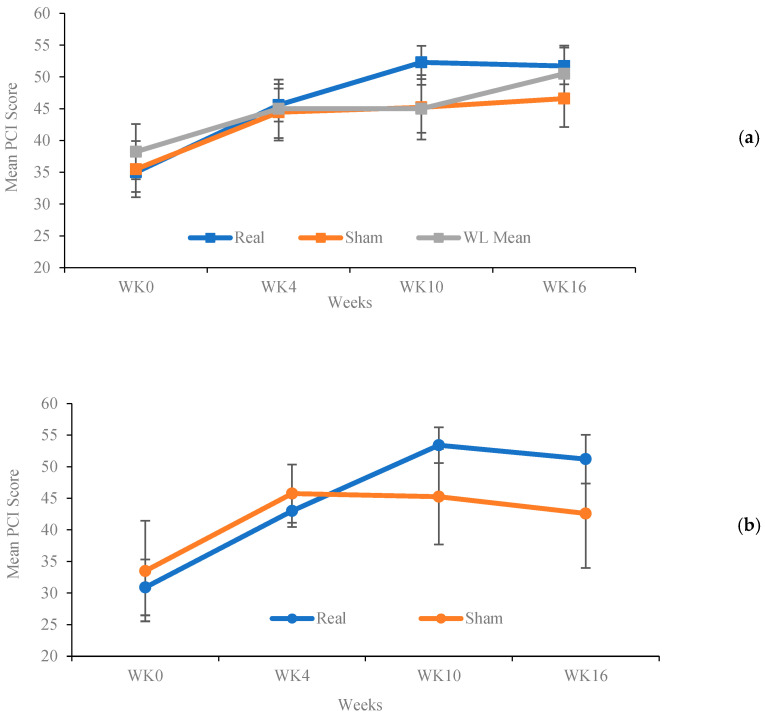
(**a**) Functional assessment of cancer therapy—cognitive function perceived cognitive impairment (FACT-Cog PCI) mean change by treatment group over time (all patients). (**b**) Functional assessment of cancer therapy—cognitive function perceived cognitive impairment (FACT-Cog PCI) mean change by treatment group over time (patients with HVLT-R T score < 50). WLC was not included because there was only one participant.

**Table 1 curroncol-32-00027-t001:** Baseline characteristics of patients.

Characteristics	No. (%)			
	Total	Real Acupuncture	Sham Acupuncture	Waitlist Control
**Mean age (SD)** (n = 32)	58.0 (11.9)	57.3 (12.6)	60.6 (7.5)	55.4 (16.3)
**Sex** (n = 32)				
Female	25 (78.1)	15 (79.0)	7 (77.8)	3 (75.0)
Male	7 (21.9)	4 (21.0)	2 (22.2)	1 (25.0)
**Race** (n = 32)				
Nonwhite	10 (31.3)	7 (36.8)	2 (22.2)	1 (25.0)
White	22 (68.7)	12 (63.2)	7 (77.8)	3 (75.0)
**Ethnicity** (n = 32)				
Hispanic or Latino	3 (9.4)	1 (5.3)	2 (22.2)	0 (0.0)
Not Hispanic or Latino	29 (90.6)	18 (94.7)	7 (77.8)	4 (100.0)
**Education** (n = 32)				
Some College	3 (9.4)	2 (10.5)	1 (11.1)	0 (0.0)
College Degree	12 (37.5)	8 (42.1)	1 (11.1)	3 (75.0)
Advanced Degree	17 (53.1)	9 (47.4)	7 (77.8)	1 (25.0)
**Cancer Type** (n = 32)				
Breast	11 (34.4)	8 (42.1)	1 (11.1)	2 (50.0)
Prostate	6 (18.8)	3 (15.8)	2 (22.2)	1 (25.0)
Gynecological	8 (25.0)	5 (26.3)	3 (33.3)	0 (0.0)
Colorectal	6 (18.8)	3 (15.8)	2 (22.2)	1 (25.0)
Bladder	1 (3.1)	0 (0.0)	1 (11.1)	0 (0.0)
**Previous Treatments** (n = 32)				
Surgery	27 (84.4)	16 (84.2)	7 (77.8)	4 (100.0)
Chemotherapy	15 (46.9)	9 (47.4)	4 (44.4)	2 (50.0)
Radiation	12 (37.5)	8 (42.1)	3 (33.3)	1 (25.0)
Reconstructive Surgery	5 (15.6)	2 (10.5)	1 (11.1)	2 (50.0)
Hormonal Therapy	4 (12.5)	3 (15.8)	1 (11.1)	0 (0.0)
Bone Marrow Transplant	1 (3.1)	0 (0.0)	0 (0.0)	1 (25.0)
Biological/Immunotherapy	1 (3.1)	1 (5.3)	0 (0.0)	0 (0.0)
Other	4 (12.5)	3 (15.8)	1 (11.1)	0 (0.0)
**Years since cancer diagnosis, mean (SD)** (n = 32)	5.7 (10.8)	6.4 (13.0)	5.3 (5.9)	3.4 (3.4)
**Cancer Stage** (n = 28)				
Stage 0	2 (7.1)	0 (0.0)	1 (12.5)	1 (25.0)
Stage I	10 (35.7)	7 (43.8)	2 (25.0)	1 (25.0)
Stage II	10 (35.7)	6 (37.5)	3 (37.5)	1 (25.0)
Stage III	6 (21.4)	3 (18.8)	2 (25.0)	1 (25.0)
**Baseline FACT-Cog PCI *, mean (SD)** (n = 32)	35.6 (13.0)	35.0 (13.5)	35.5 (13.3)	38.3 (8.7)

* Abbreviation: FACT-Cog PCI: Functional Assessment of Cancer Therapy—Cognitive Function Perceived Cognitive Impairment.

**Table 2 curroncol-32-00027-t002:** Functional assessment of cancer therapy—cognitive function perceived cognitive impairment (FACT-Cog PCI) mean change from baseline by treatment group and between-group differences.

	Real Acupuncture (RA)			Sham Acupuncture (SA)			Waitlist Control (WLC)	Difference Between RA and SA in Change from Baseline (95% CI)	*p*-Value (RA vs. SA)
	Mean Change from Baseline (95% CI)	Difference from WLC in Change from Baseline (95% CI)	*p*-Value (RA vs. WLC)	Mean Change from Baseline (95% CI)	Difference from WLC in Change from Baseline (95% CI)	*p*-Value (SA vs. WLC)	Mean Change from Baseline (95% CI)		
**All Patients**									
Week 10	17.31 (12.53 to 22.09)	10.56 (−0.89 to 22.00)	0.074	9.72 (2.78 to 16.66)	2.97 (−9.53 to 15.47)	0.64	6.75 (−3.66 to 17.16)	7.59 (−0.84 to 16.02)	0.081
Week 16	17.34 (12.48 to 22.20)	5.08 (−6.40 to 16.57)	0.39	11.09 (4.15 to 18.03)	−1.16 (−13.66 to 11.35)	0.86	12.25 (1.84 to 22.66)	6.24 (−2.22 to 14.71)	0.15
**Patients with Baseline HVLT-R** *** T score <50**									
Week 10	20.41 (13.55 to 27.27)	NA *	NA	11.05 (0.56 to 21.54)	NA	NA	NA	9.35 (−3.08 to 21.78)	0.15
Week 16	20.65 (14.02 to 27.27)	NA	NA	7.28 (−3.30 to 17.86)	NA	NA	NA	13.36 (0.86 to 25.86)	0.044

* Abbreviation: HVLT-R: Hopkins Verbal Learning Test—Revised. Since only one patient in the WLC met the criteria of HVLT-R T < 50, we did not conduct a comparison analysis with the WLC group.

**Table 3 curroncol-32-00027-t003:** Hopkins Verbal Learning Test—Revised (HVLT-R) mean change from baseline by treatment group (all patients).

	Real Acupuncture		Sham Acupuncture		Waitlist Control	
HVLT-R Total Recall	N	Mean (SD)	N	Mean (SD)	N	Mean (SD)
Baseline	19	47.10 (10.46)	9	47.11 (14.56)	4	50.75 (9.03)
Week 10	19	44.58 (10.92)	8	47.75 (14.07)	4	50.00 (12.27)
Week 16	18	49.06 (7.44)	8	44.75 (14.90)	4	53.75 (14.08)
**HVLT-R Delayed Recall**						
Baseline	19	49.42 (9.63)	9	46.33 (12.79)	4	48.00 (14.07)
Week 10	19	46.16 (11.49)	8	45.00 (10.17)	4	56.25 (5.50)
Week 16	18	50.00 (11.15)	8	45.12 (12.98)	4	39.75 (20.82)

**Table 4 curroncol-32-00027-t004:** Hopkins Verbal Learning Test—Revised (HVLT-R) mean change from baseline by treatment group (patients with HVLT-R T score < 50).

	Real Acupuncture		Sham Acupuncture	
HVLT-R Total Recall	N	Mean (SD)	N	Mean (SD)
Baseline	10	39.30 (7.47)	4	43.70 (8.67)
Week 10	10	41.40 (10.68)	4	49.25 (9.43)
Week 16	10	45.20 (5.63)	4	33.75 (10.28)
**HVLT-R Delayed Recall**				
Baseline	10	43.70 (8.67)	4	39.25 (13.57)
Week 10	10	42.70 (10.94)	4	44.50 (8.18)
Week 16	10	44.20 (9.64)	4	35.25 (10.18)

## Data Availability

The data of this study will be available from the corresponding author upon reasonable request.

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
