# Peer review of "Feasibility and Preliminary Effects of Acupuncture for Cognitive Dysfunction in Diverse Cancer Survivors: A Pilot, Randomized, Placebo-Controlled Trial"

_curroncol, 2025, doi:10.3390/curroncol32010027_

Round 1

Reviewer 1 Report

Comments and Suggestions for Authors

great study with important topic utilizing acupuncture

several revisions are required for below

1. abstract: clarify number of control and intervention, also this sentence is hard to understand, pls clarify and revise more easy to understand, what are the subgroups: The improvements from RA persisted through week 16 in both the total and sub-groups. 

2. introduction: Therefore, more studies with rigorous design are needed to 62 investigate the efficacy of acupuncture on CRCD in a broader and more diverse cancer 63 population compared to sham controls

authors mentioned this, but at this time, the presentation is not focusing on diverse cancer patients, but patients population including 30% are minorities look good

I would strongly recommend to change the manuscript 

1) title: add in various and diverse racial/ethnic cancer 

2) intro: add paragraph- CRCI intervention is predominantly done in White, so further research is needed to consider racial/ethnic minorities .. add the paragarph to support this. 

3) discussion: same thing, add some paragraph regarding CRCI and minority cancer patients, require further support to manage symptoms. 

**methods

what are the covaraites, define and add covariate adjust/unadjusted results

for example, types of cancer DIAGNOSES, cancer stages, types of cancer treatments should be adjusted as they are associated with CRCI. 

Discussion: clinical implication 

supplementary file: author still left the comments, please delete all comments boxes. 

Reviewer 2 Report

Comments and Suggestions for Authors

I appreciate the thoroughness of this study of real acupuncture (RA) and the use of sham acupuncture (SA) as a control, instead of merely having a waitlist control (WLC). Moreover, I also like that you discuss MID (minimal clinically important differences), instead of just statistical significance. Most of the manuscript is well written but I have some major comments:

1.      I suggest you are more distinct with your Aims: Is feasibility a primary outcome measure? And possible clinical effects a secondary measure?

2.      The study is named as a feasibility study (which I agree) but it focuses also on “preliminary effects”. I can understand that you want to present clinical data and you choose suitable instruments.
However, if the clinical effect is one of your endpoints, you need to present a power calculation and design your study accordingly. This is not done (no formal power calculation or reasoning). However, you mention that you aimed at 80 participants (how did you arrive at this conclusion?) but included only 32.
Thus, it is very hard to show preliminary effects in a study that is obviously under-powered.
You mention all this, but I suggest you are even more critical in your discussion/ limitations.

3.      Obviously, the study is underpowered for statistical comparisons. This becomes clear considering that there are no statistically significant differences, and the 95%CI are very broad and overlapping.
Also in this sense, I would recommend a more critical writing (e.g., that the data are hypothesis-generating, rather than suggesting that there are clinical differences. There may very well be clinical differences, but you cannot draw that conclusion from a non-conclusive study.
As an example, you mention that the difference for RA is 17.3, which is well above the expected MID of 7.5. However, also the difference for sham acupuncture, 9.2 is above 7.5 and even the WLC group “improved” with a difference of 6.8, i.e., near the level of a MID of 7.5. This is the reason why you should be conservative when discussing an underpowered study.

4.      The study went on for 10 weeks. As a reader I am curious about the data at 4 weeks: where there any improvements? Or do you need 10 weeks (at least) to improve?

5.      I also recommend you to be a bit more critical as regards your sampling which is not random (as regards cancer): there is an obvious overrepresentation of highly educated, women and the patients are much younger than the typical cancer population (mean age 58 years and recruited 5.7 years after diagnosis, meaning a mean age of 52 years at diagnosis – extremely low for persons with prostate cancer).
Data can only be generalized to similar contexts.

6.      As the main focus is on feasibility, I would like to know more about those excluded because of being “not eligible” or the group “future potential”. Why was not this study design appropriate (feasible) for these patients?

7.      I have some concern about the recruitment: 32 participants recruited during a period of 6 months, ie., 5 persons per month or roughly 1 patient per week. If the study is feasible, why was the recruitment so slow?

8.      Phase I trial: In the discussion, you label your study as a Phase 1 study. If this is the case, this should be stated in the Methods section (and in a Phase 1 study feasibility and safety should be in the focus).

9.      I find it surprising that “years since cancer diagnosis” were 5.7 years. This is very long after primary treatment. Why is it that you did not find patients that had recently finished their treatment (as symptoms of “chemo brain” often are most obvious early in the course)?

10.  In conclusion, as feasibility and safety seem to be primary outcome measures, I suggest you focus a bit more on those aspects (now the preliminary effects get a lot of space).

Although I have posed several questions, I appreciate your effort to perform a randomized study with a reliable control group (sham acupuncture).

Round 2

Reviewer 1 Report

Comments and Suggestions for Authors

great work 

Reviewer 2 Report

Comments and Suggestions for Authors

Dear authors

The revised manuscript is much improved and I have no further comments.